# The Association between Bone Mineral Density and Periodontal Disease in Middle-Aged Adults

**DOI:** 10.3390/ijerph18063321

**Published:** 2021-03-23

**Authors:** Hsin-Hua Chou, Sao-Lun Lu, Sen-Te Wang, Ting-Hsuan Huang, Sam Li-Sheng Chen

**Affiliations:** 1School of Oral Hygiene, College of Oral Medicine, Taipei Medical University, Taipei 11031, Taiwan; hhchou@tmu.edu.tw; 2Dental Department of Wan-Fang Hospital, Taipei Medical University, Taipei 11600, Taiwan; 3School of Dentistry, College of Oral Medicine, Taipei Medical University, Taipei 11031, Taiwan; ddaniellu@hotmail.com (S.-L.L.); u9718301@cmu.edu.tw (T.-H.H.); 4Department of Family Medicine, School of Medicine, College of Medicine, Taipei Medical University, Taipei 11031, Taiwan; wangader@gmail.com; 5Department of Family Medicine, Taipei Medical University Hospital, Taipei 11031, Taiwan

**Keywords:** bone mineral density, periodontal disease, osteoporosis, risk factor

## Abstract

The association between osteoporosis and periodontal disease (PD) has been revealed by previous studies, but there have been few studies on the association in younger adults. We enrolled a total of 7298 adults aged 40 to 44 who underwent PD screening between 2003 and 2008. Data on quantitative ultrasound for the measurement of bone mineral density (BMD) were collected for the diagnostic criteria of osteopenia and osteoporosis. The Community Periodontal Index (CPI) was measured for defining PD. A multiple logistic regression model was used to assess the effect of low bone mass on the risk of PD. Of 7298 enrollees, 31% had periodontal pockets >3 mm, 36.2% had osteopenia, and 2.1% had osteoporosis. The 39.8% of PD prevalence was high in adults with osteoporosis, followed by 33.3% in osteopenia. A negative association was found between BMD and CPI value (*p* < 0.0001). Low bone mass was associated with the risk of PD (adjusted OR: 1.13; 95% CI:1.02–1.26) after adjusting the confounding factors, including age, gender, education level, overweight, smoking status, past history of osteoporosis, and diabetes mellitus. An association between BMD and PD among young adults was found. An intervention program for the prevention of PD and osteoporosis could be considered starting in young adults.

## 1. Introduction

Osteoporosis is a systemic skeletal condition characterized by low bone density and the deteriorated microarchitecture of bone tissue, and is mainly seen in the elderly population. It is more common in women than in men. This condition is caused by physiologic age-related bone loss occurring after menopause. More than one third of the female population aged over 65 suffers from signs and symptoms of osteoporotic fractures [1,2].

Osteoporosis is common in the elderly and there is a relatively low prevalence of osteoporosis in young adults [3]. Wowern et al. revealed that severe osteoporosis which significantly reduces the bone mineral content of the jaws may be associated with a less favorable attachment level in the case of periodontal disease (PD) [4]. Since the loss of alveolar bone is a prominent feature of PD, severe osteoporosis is suspected of being an aggravating factor in the case of PD.

The Global Burden of Disease Study reported that severe PD was the 11th most prevalent condition in the world in 2016 [5,6]. The prevalence of PD was reported to be between 20% and 50% globally [7]. The overall periodontitis prevalence has shown to increase with age, and the incidence rises sharply in adults aged 30–40 years [8].

To our knowledge, the relationship between osteoporosis and PD in young adults is rarely addressed and difficult to establish due to the association being confounded by other factors such as gender, hormone intake, smoking, race, and age [9]. Furthermore, previous studies have assessed osteoporosis from changes in the metacarpal index [10,11], which may not give a correct estimate of the bone mineral content of the jaws.

In Taiwan, population-based PD screening has been conducted since 2003 among residents aged 35–44 years, in which information on both demographic and chronic disease has been documented [12]. This would enable us to investigate osteoporosis and PD among young adults with other confounding factors.

We therefore aim to investigate the association between osteoporosis and periodontal disease in the context of the current periodontal etiology model in young adults based on data from a large population-based periodontal disease screening.

## 2. Materials and Methods

### 2.1. Setting and Study Population

A population-based cross-sectional study design was adopted, with a total of 15,537 subjects aged 35–44 years from between 2003 and 2008 being recruited from the Keelung community-based integrated screening program (CIS), Taiwan. Briefly, CIS is an integrated disease screening program for adults aged over 30 years old from Keelung City, Taiwan. The main screening items consist of cancers (oral, breast, liver, cervical, and colorectal cancer) and chronical diseases (hypertension, hyperglycemia, and hyperlipidemia) in light of evidence-based medicine. The assessment of PD for individuals aged 35–44 years has been incorporated into the program since 2003.The details of CIS with PD screening have been described in detail elsewhere [12]. This study was approved by the Joint Institutional Review Board of Taipei Medical University (TMU-JIRB No.201207011).

### 2.2. Periodontal Survey Procedures

The periodontal examination by the Community Periodontal Index (CPI) (WHO, 1997) has been used to identify the status of PD. The examiner used a WHO probe to obtain five levels of ordinal scores (healthy, gingival bleeding, calculus, shallow pockets of 4–5 mm, and deep pockets of 6 mm or deeper). CPI scores by six sextants including two anterior sextants and four posterior sextants for each individual were collected in this study. To be efficient for the large scale survey in the community, WHO suggests the examination for index teeth (1–2 teeth) from each sextant to represent the periodontal status of that sextant. The highest CPI score among the six sextants was adopted to represent the periodontal status for each individual [13].

### 2.3. Questionnaire Survey

The structured questionnaire contained items on (1) lifestyle, with details of cigarette smoking, alcohol consumption, and betel-quid chewing, as well as the associated frequency of consumption and duration of habits; (2) personal disease history of osteoporosis, diabetes mellitus, hypertension, CVD, and cerebrovascular disease. The questionnaire was administered by public health nurses in the CIS program.

### 2.4. The Measurement of Metabolic Syndrome (MetS)

MetS was defined according to the joint scientific statement criteria [14], which requires the presence of 3 or more of the following criteria: (1) elevated fasting glucose value (≥100 mg/dL); (2) elevated BP (systolic pressure ≥ 130 or diastolic pressure ≥ 85 mm Hg); (3) reduced HDL cholesterol level (<40 mg/dl in men or <50 mg/dL in women); (4) elevated triglyceride level (≥150 mg/dL); and (5) elevated waist circumference (≥90 cm in men or ≥80 cm in women).

### 2.5. The Measurement of Bone Mineral Density

The quantitative ultrasound measurements were performed at one calcaneus for each participant aged over 40 by well-trained technicians to determine the individuals’ bone mass. T-score was recorded after quantitative ultrasound assessment. Bone mineral density (BMD) values are expressed as absolute values in g/cm^2^ or as T-scores from the young adults. In the current analysis, osteopenia was defined as a T-score between −1.0 and −2.5, while osteoporosis was defined as a T-score < −2.5, as recommended by the World Health Organization guidelines [15].

### 2.6. Statistical Analyses

The participants aged between 40 and 44 years attending both BMD measurement and periodontal disease screening programs were recruited in our analysis. A total of 7298 samples with complete BMD information were used to estimate the severity of PD. The dependent variable in our study was PD. Subjects with CPI scores over 3 was defined as PD. Four sets of variables were tested collectively against PD in multivariate logistic regression modeling: (1) osteoporosis defined by BMD, (2) demographic and behavioral variables commonly included in interview-based surveys (including age, gender, past history of osteoporosis, smoking, betel-quit chewing, and alcohol consumption), (3) overweight (a BMI greater than or equal to 25) by the WHO’s definition, and (4) variables addressing the metabolic syndrome status. The statistical significance level was set at 5%. The ANOVA test was used to test the differences between CPI score and BMD value. The significant risk factors were selected in a multivariate logistic regression model based on the stepwise method. All the analyses were performed with the SAS statistical software, version 9.4 (SAS institute, Inc., Cary, NC, USA).

## 3. Results

Among the 7298 subjects, the overall prevalence of PD (CPI ≥ 3) was 31%. The prevalence of PD increased with age from 27.5% for age 40 years to 33.9% for age 44 years (Table 1). The prevalence of PD was 37.9% higher in males than in females (27.5%). The prevalence of PD increased with education levels from high (26.9%) to low (35.1%). PD prevalence was highest in subjects with osteoporosis (39.9%), followed by subjects with osteopenia (33.3%), compared to those without low bone mass (29.5%). The prevalence of osteoporosis (BMD ≤ −2.5) was 2.1% (153/7298). The prevalence of osteopenia (−2.5 < BMD ≤ −1) was 36.2% (2644/7298).

The distribution of BMD by CPI score is presented in Table 2. The higher the CPI value was, the more negative the mean value of BMD was. These differences were statistically significant (*p* < 0.0001) when examined using ANOVA. The risk factors in association with the risk of PD are shown in Table 3. In the univariate analysis, the presence of low BMD (BMD < −1, osteopenia or osteoporosis) was associated with an increase in the risk of developing PD of 21% (OR: 1.21, 95% CI: 1.10–1.34). Other significant risk factors included age, male, low education level, smoker, past history of osteoporosis, overweight, diabetes mellitus, and metabolic syndrome.

After controlling for these significant confounding factors, low bone mass (BMD < −1) was still associated with the risk of PD (adjusted odds ratio (aOR) = 1.13, 95% CI: 1.02–1.26).

## 4. Discussion

Despite the fact that osteoporosis and PD are prevalent among elderly populations, the PD incidence increased dramatically in young adults. In addition, evidence of the association between osteoporosis and PD has not been well documented in young adults. We investigated the association between low bone mass and PD based on a community-based epidemiological study of PD in Keelung, Taiwan.

We found that there is a significance correlation between BMD and CPI in that the lower the mean BMD, the higher the CPI. The effect of osteopenia or osteoporosis on the PD was further supported by multiple logistic regression analysis. After adjusting for well-known confounding factors including age, gender, education level, smoking, overweight, and either diabetes or MetS, low bone mass was associated with a 13% increase in the risk of PD (aOR = 1.13; 95% CI: 1.02–1.26). Our result is consistent with that of the previous cross-sectional studies covering with younger adults, showing that osteoporosis can be considered a risk factor for PD [16,17]. A meta-analysis also reported that subjects with osteoporosis were 2.17 (OR, 2.17; 95% CI, 1.80–2.61) times more likely to have periodontitis than those who are free of osteoporosis but the analysis included the studies with elder people [18].

The relationship between osteoporosis and PD may be supported by three mechanisms that have been mentioned, comprising systemic to local bone resorptive disease, hormonal impact on bone homeostasis inflammation, and inflammation and bone homeostasis [19].

Aside from the hormonal effect, the systemic to local bone resorptive disease is considered for describing the link between osteoporosis and PD. A robust association between systematic and local osteoporotic changes in the jaw has been reported. Osteoporosis of the alveolar bone may lower the resistance of the periodontium to infectious challenge and may result in a local infection of the periodontium that first invades the cortical bone and results in a dimensional change in the alveolar ridge [20].

However, a non-infectious mechanism was also considered to be a possible mechanism, since the association between alveolar bone resorption and tooth loss has been shown to be stronger than other clinical measurements of PD [21].

Another possible mechanism involves inflammatory cytokines, in that osteoporosis patients have elevated systemic levels of pro-inflammatory cytokines IL-1, IL-6, and TNF-α [20]. These cytokines account for osteoclastogenic bone resorption-inducing cytokines and are also involved in the tissue response to PD [22].

Moreover, homeostatic bone remodeling involves “physiologic inflammation” to recruit non-phlogistic macrophages for the clearance of apoptotic bone cells [23,24]. Interestingly pro-resolving lipid mediators (SPMs) have been unveiled as contributors to an active process of inflammation and are involved in bone homeostasis. Therefore, it is encouraging to understand the role of SPMs in bone homeostasis to disentangle the links between osteoporosis, inflammation, and periodontitis [25].

It is inevitable that there are also studies that have shown a lack of association between osteoporosis and PD [20,26]. Moreover, there is a study that demonstrated that patients with periodontal treatment experience an increased risk of osteoporosis [27]. On the other hand, the inverse relationship between past history of treated osteoporosis and periodontal disease was elucidated in our finding. As a result, whether there is causation between the two conditions is still elusive, in terms of whether osteoporosis causes PD or vice versa. More studies are required. However, our study was based on a large community-based data set and targeted young adults. We believe that our result could be used as a warning to raise public awareness about osteoporosis and PD in young adults.

According to Nordin’s definition, osteoporosis is present when the concentration of bone lies more than 2 standard deviations below the mean of young normal values of the same sex [28]. The World Health Organization has also defined osteoporosis as a BMD < 2.5 SD below the mean value of peak bone mass in young normal women [15]. Someone might argue that the dichotomizing subjects by BMD threshold values might result in misclassification bias and loss of information. In this study, both of a continuous variable and a dichotomized variable for BMD were used for the statistical analysis. All these analyses lead to a same result, as demonstrated by a current study of osteoporosis and PD.

Several limitations exist in our study. First, the probing depth was used as a dependent variable to represent PD severity in our study. However, probing depth readings probably do not offer a good measurement of PD history [29]. There are also a few uncontrolled studies that have found an inverse relationship, concluding that subjects with higher BMD retain their teeth with deeper probing depths [30]. Second, the analysis excluded treatment information for periodontal disease which might affect the relationship between bone mineral density and periodontal disease. However, as our National Health Insurance system included the comprehensive periodontal treatment project in 2010, the proportion of treatment for periodontal disease by asymptomatic subjects receiving periodontal disease screening during the study period was expected to be very low. Third, there were still other confounder factors which we could not take into account—for example, vitamin D. However, we believe that the effect from this would be minimal because we focused on young adults. Third, as our design is a cross-sectional study, we could not elucidate the causation between osteoporosis and PD.

## 5. Conclusions

In conclusion, based on community data, we demonstrated that low bone mass increased the risk of PD among young adults. An intervention program for the prevention of PD and osteopenia or osteoporosis may be considered for young adults.

## Figures and Tables

**Table 1 ijerph-18-03321-t001:** Baseline characteristics of study subjects.

Variables	CPI < 3	%	CPI ≥ 3	%	Total	*p*
Age						<0.0001
40	926	72.5	351	27.5	1277	
41	1033	72.4	393	27.6	1426	
42	1037	67.8	492	32.2	1529	
43	1005	66.4	508	33.6	1513	
44	1026	66.1	527	33.9	1553	
Gender						<0.0001
Female	3439	72.5	1303	27.5	4742	
Male	1588	62.1	968	37.9	2556	
Education level						
Junior high school or below	1484	73.1	546	26.9	2030	<0.0001
Senior high school	2428	68.2	1130	31.8	3558	
College or above	1089	64.9	589	35.1	1678	
Bone Mineral Density (BMD)						0.0002
BMD > −1	3172	70.5	1329	29.5	4501	
−2.5 < BMD ≤ −1 (osteopenia)	1763	66.7	881	33.3	2644	
BMD ≤ −2.5 (osteoporosis)	92	60.1	61	39.9	153	
Past History of Osteoposis						
No	4757	68.6	2180	31.4	6977	0.0089
Yes, without medications	172	71.7	68	28.3	240	
Yes, with medications	98	81.0	23	19.0	121	
BMI						<0.0001
18 ≤ BMI < 25	4118	70.0	1767	30.0	5885	
BMI ≥ 25	887	63.8	503	36.2	1390	
Smoking						<0.0001
No	3652	72.2	1406	27.8	5058	
Yes	1283	60.4	841	39.6	2124	
Alcohol Drinking						
	3747	71.1	1525	28.9	5272	<0.0001
No	1162	62.1	709	37.9	1871	
Betel-quid Chewing						
No	4570	70.1	1854	29.9	6524	<0.0001
Yes	347	54.8	286	45.2	633	
Diabetes Mellitus						<0.0001
No	4883	69.4	2157	30.6	7040	
Yes	132	55.2	107	44.8	239	
Component of Metabolic Syndrome						
Central Obesity						<0.0001
No	3992	70.7	1654	29.3	5646	
Yes	1035	62.6	617	37.4	1652	
Hypertriglyceridemia						<0.0001
No	3859	70.3	1630	29.7	5489	
Yes	1168	64.6	641	35.4	1809	
Low HDL-C						<0.0001
No	3692	70.8	1335	29.2	5214	
Yes	1522	64.1	749	35.9	2084	
Elevated Blood Pressure						0.0029
No	4178	69.6	1822	30.4	6000	
Yes	849	65.4	449	34.6	1298	
Hyperglycemia						0.0012
No	4415	69.6	1932	30.4	6347	
Yes	612	64.4	339	35.6	951	
Metabolic Syndrome						<0.0001
No	4270	70.4	1799	29.6	6069	
Yes	757	61.6	472	38.4	1229	

**Table 2 ijerph-18-03321-t002:** The distribution of bone mineral density by the community periodontal index.

CPI	N	Bone Mineral Density			
		Mean	SD	Min	Max
0	483	−0.5459	0.96	−3.33	6.40
1	997	−0.5635	1.02	−4.90	3.40
2	3547	−0.6083	1.04	−5.81	4.80
3	1845	−0.7029	1.01	−4.48	4.70
4	426	−0.7433	1.00	−3.10	3.70

ANOVA test: F = 5.93, *p* < 0.0001.

**Table 3 ijerph-18-03321-t003:** The association between low bone mass, other factors, and periodontal disease (PD).

Variables	Periodontal Disease (CPI ≥ 3)			
	OR	95% CI	aOR	95% CI
Age	1.09	1.06–1.13	1.09	1.05–1.13
Male vs. Female	1.61	1.45–1.78	1.39	1.21–1.58
Education Level				
Junior high school or below vs. College or above	1.27	1.13–1.44	1.30	1.15–1.48
Junior high school or below vs. College or above	1.48	1.29–1.71	1.51	1.30–1.76
BMD < −1 vs. BMD ≥ −1	1.21	1.10–1.34	1.13	1.02–1.26
History of Osteoposis				
Without medications vs. non-osteoposis	0.86	0.65–1.15	0.86	0.64–1.15
With medications vs. non-osteoposis	0.51	0.32–0.81	0.47	0.30–0.74
Smoking (Yes vs. No)	1.70	1.53–1.89	1.36	1.19–1.56
Alcohol Drinking (Yes vs. No)	1.50	1.34–1.68		
Betel-quid Chewing (Yes vs. No)	1.93	1.63–2.27		
BMI ≥ 25 vs. 18 ≤ BMI < 25	1.32	1.17–1.49	1.15	1.01–1.30
Diabetes Mellitus	1.84	1.42–2.38	1.62	1.24–2.13
Metabolic Syndrome	1.48	1.30–1.68	1.21 *	1.05–1.41

* Adjusted in a separate multivariate model.

## Data Availability

The data that support the findings of this study are available on request from the corresponding author. The data are not publicly available due to privacy of research participants.

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
