# Peer review of "The Association between Bone Mineral Density and Periodontal Disease in Middle-Aged Adults"

_ijerph, 2021, doi:10.3390/ijerph18063321_

Round 1

Reviewer 1 Report

Confounding factors are always an issue. 

  1. For that reason, I recommend a MaxR or other multiple regression statistical test.
  2. Whether the osteopenia/osteoporosis was under treatment must also be considered.
  3. The effect of medications must be considered.  Relationship of periodontal disease to medications in general and specifically those used to treat osteoporosis/osteopenia must be considered

Author Response

Reviewer 1

  1. Confounding factors are always an issue. For that reason, I recommend a MaxR or other multiple regression statistical test.

Ans: Agree. The stepwise selection method was applied to identify covariables for inclusion in logistic regression models.

  1. Whether the osteopenia/osteoporosis was under treatment must also be considered.

Ans: Agree. The proportion of prescribed medication used by asymptomatic subjects receiving bone mass measurement could be expected to be low. In this study, self-reported history of osteoporosis at baseline was 4.95% (361/7298). Among subjects with past history of osteoporosis, only 121 (36.1%) treated with medications. Most of asymptomatic screened subjects with osteopenia or osteoporosis would be under treatment. This point has been elucidated in Results and addressed in Discussion. (Page 4, Line 7; Page 7, Line 29-31)

  1. The effect of medications must be considered. Relationship of periodontal disease to medications in general and specifically those used to treat osteoporosis/osteopenia must be considered.

Ans: Thank you for the valuable commend. The history of osteoporosis with medications has been considered in our analysis. The inverse relationship between past history of treated osteoporosis and periodontal disease were found. The low bone mass was still associated with the risk of periodontal disease after adjusting these covariates. However, the analysis excluded treatment information for periodontal disease which could affect the relationship between bone mineral density and periodontal disease. However, as our National Health Insurance system included the comprehensive periodontal treatment project in 2010, the proportion of treatment for periodontal disease by asymptomatic subjects receiving periodontal disease screening during the study period was expected to be very low. This limitation has also been addressed in Discussion. (Page 8, Line 5-11)

Reviewer 2 Report

The authors conducted a cross-sectional study to investigate the association between bone mineral density and periodontal disease in the middle-age adults aged between 40 and 44 years old. This provides the appropriate evidence to support the hypothesis that bone mineral density and periodontal disease and this manuscript was written with well structured and easy read. Moreover, this study extended this association into the younger adult population. However, several minor issues need the authors to clarify.

  1. The obesity definition was 25 kg/m2. This criteria seems not the official criteria of obesity. Please have a statement to explain about this cut-point.
  2. In the study population section, the authors described "PD screening has been conducted since 2003 among 50 residents aged 35-44 years" . However, the tables and abstract presented the adults only aged between 40- 44 years old. Please the authors explain it.
  3. The betel-quid chewing should affect the PD and alcohol drinker [pmid=10780863] is one of the risk factors of osteoporosis. These two risk factors should be controlled in the final multivariable model.
  4. Table 2. The bone mineral density is continued variable and the Periodontal Disease index was ordinal (categorical) variable. The general linear model or variance analysis may be more appropriate.

Author Response

Reviewer 2

The authors conducted a cross-sectional study to investigate the association between bone mineral density and periodontal disease in the middle-age adults aged between 40 and 44 years old. This provides the appropriate evidence to support the hypothesis that bone mineral density and periodontal disease and this manuscript was written with well structured and easy read. Moreover, this study extended this association into the younger adult population. However, several minor issues need the authors to clarify.

  1. The obesity definition was 25 kg/m2. This criteria seems not the official criteria of obesity. Please have a statement to explain about this cut-point.

Ans: Thank you for the valuable comment. The WHO regards a BMI greater than 25 is considered overweight and above 30 is considered obese. BMI at 25 of cut-off was used as overweight in our analysis. We have used overweight (a BMI greater than or equal to 25) throughout the manuscript instead of obesity.

  1. In the study population section, the authors described "PD screening has been conducted since 2003 among 50 residents aged 35-44 years" . However, the tables and abstract presented the adults only aged between 40- 44 years old. Please the authors explain it.

Ans: The quantitative ultrasound measurements were performed at one calcaneus for each participant aged over 40. Therefore, the adults aged between 40-44 years old were enrolled in our analysis. This point has been addressed in the Method section. (Page 3, Line 12 and Line 21-23)

  1. The betel-quid chewing should affect the PD and alcohol drinker [pmid=10780863] is one of the risk factors of osteoporosis. These two risk factors should be controlled in the final multivariable model.

Ans: Agreed. The two factors including betel-quid chewing and alcohol intake have been included in our analysis. However, the two risk factors did not include in the final multivariable model by model selection.

  1. Table 2. The bone mineral density is continued variable and the Periodontal Disease index was ordinal (categorical) variable. The general linear model or variance analysis may be more appropriate.

Ans: Thank you for your valuable suggestion. The differences of mean BMD have been examined using ANOVA test. The result presents in Table 2. 

Round 2

Reviewer 1 Report

It could be improved by specific identification of medications consumed that affect osteoporosis (e.g., prilosec)

Reviewer 2 Report

I have no further questions.